# Numerical Simulation and Optimization of Microwave Heating Effect on Coal Seam Permeability Enhancement

Ali Jebelli [1], Arezoo Mahabadi [2] and Rafiq Ahmad [1,*]

1    Department of Mechanical Engineering, University of Alberta, Edmonton, AB T6G 1H9, Canada; jebelli@ualberta.ca
2    Department of Engineering Sciences, University of Tehran, Tehran 1417935840, Iran; a.mah.abadi@ut.ac.ir
*    Correspondence: rafiq.ahmad@ualberta.ca

**Abstract:** In coal mining operations, coalbed methane is one of the potential hazards that must be extracted to prevent an explosion of the accumulated gas and environmental pollution. One of the mechanisms is using microwave irradiation so that the thermal stress caused by microwave heating generates fractures. In this research, we investigated the most important parameters affecting the electric and thermal fields' distribution in coal in order to identify the effective parameters that achieve the highest temperature increase rate and to reach the highest impact and efficiency of the system with the least amount of consumed energy. In this paper, using Maxwell equations, heat transfer, mass transfer and coupling them by COMSOL, we have simulated the radiation of electromagnetic field and heat in the cavity and coal, and we have also shown the temperature dispersion inside the coal. The parameters studied included the amount of coal moisture (type of coal), operating frequency, input power and heating time, location of the waveguide, the size of the waveguide and the location of the coal, and finally the parameters were re-examined in a secondary standard cavity to separate the parameters related to the size of the environment and the cavity from the independent parameters. The results of this study show that the most effective parameter on the electric and thermal fields' distribution within coal is the size of the resonance chamber. Additionally, the results show that the moisture of 5%, the highest input power and cutoff frequency close to the operating frequency cause the highest average temperature inside the coal, but many parameters such as operating frequency, waveguide location and coal location should be selected depending on the chamber size.

**Keywords:** COMSOL; coal microwave heating; coal permeability enhancement; cavity; coalbed methane

## 1. Introduction

Methane in coal seams in the past was known as a risk factor but today as an economical and extractable commodity [1]. Coalbed methane (CBM) is an unconventional form of natural gas that forms biogenically or thermogenically in coal deposits or coal seams. Biogenic methane gas is produced by bacteria inside the coal, but thermogenic gas is formed by chemical devolatilization of coal under pressure and heat, and methane gas is stored as adsorption in coal. Therefore, permeability is by far the most important feature controlling the gas flow in coal resources and the second important feature is the gas saturation level, which is determined by measuring the desorption and adsorption [1].

It is crucial to extract methane gas before and during underground mining operations to prevent methane eruptions and accidents caused by coalbed methane (CBM) explosions in coal. However, maximizing the extraction process while minimizing the associated pollution is one of the most challenging issues this industry is facing [2].

CBM extraction is often associated with significant water extraction, the extraction of which reduces groundwater resources and its disposal causes environmental pollution. In

general, the CBM production profile has three stages of dewatering, stable production and decline, which is different from natural gas sources in that at the beginning of the extraction process, the highest amount of methane gas and the lowest amount of water are extracted, and the gas extraction rate decreases and the water extraction rate increases.

In CBM sources, there is the highest water extraction rate in the dewatering stage, which decreases over time, and the highest methane gas extraction rate is in the stable production stage [1]. Hydraulic fracturing is one of the most basic, most commonly used methods of CBM extraction, which by pumping water into the well, methane gas along with water is removed from the coal layers, the volume of water produced according to the size of the pump and the amount of water adjustment by the operator is very variable and the produced water is either re-injected deeper or filtered and disposed on the ground [3].

Most coal seams belong to low permeability reservoirs, where gas is not produced economically without stimulating the reservoir and increasing permeability. Stimulation of seams in coal and CBM extraction are carried out in different ways. These methods are divided into three general categories of mechanical, thermal and chemical mechanisms [4], of which hydraulic techniques are part of mechanical methods. Because hydraulic methods have the possibility of water-block in micropores, groundwater contamination and gas leakage, other methods such as gas injection ($CO_2$-ECBM and $N_2$-ECBM), liquid $CO_2$ injection, liquid $N_2$ injection, high-voltage electrical fracturing and acid fracturing are considered for CBM extraction. Using a gas injection method has advantages such as storing $CO_2$ gas underground, but there is a possibility of $CO_2$ gas leakage, impossibility to control the amount of fracturing and the need for an environment with suitable permeability to prevent gas blockage.

Injection of nitrogen liquid freezes the water inside the coal pores and causes fractures due to the increased pressure in it by increasing the volume of water ice and nitrogen gasification, but liquid nitrogen and its maintenance equipment is very expensive. In the high-voltage electrical fracturing method, cracks are created in the coal by using electric shock and high heat, which have protection problems due to the use of very high voltage, and also cause huge breakage and deformation in the coalbed [4].

Using acid fracturing also requires advanced equipment and the possibility of acid reaction and its dissolution in groundwater are among the risks of using this method. Therefore, a reliable CBM extraction method should be able to control the amount of fractures in coal, its efficiency should be stable, the maintenance costs of parts should be cost effective, groundwater resources should not be damaged, it should not be possible to block the gas route and it should have high safety.

One of the new methods of methane extraction is the use of microwave heating. To study this method, it is necessary to examine all the effective parameters for designing suitable and reliable CBM extraction systems.

One of the most recent approaches for extracting this gas is the use of microwave radiations. Microwave radiations are widely used in coal processing fields such as drying, coking, pyrolysis, flotation, increasing grindability and magnetic removal [2]. Microwaves are a type of electromagnetic waves, with frequencies between 300 MHz and 300 GHz, widely used in industrial, scientific, medical and instrumental applications. By increasing the permeability of coal, such waves affect the molecules and raise the object temperature by increasing the internal vibration of molecular bonds, lengthening these bonds and enhancing the mobility of molecules and their internal energy [5]. Thus, the moisture in the coal evaporates under microwave radiation, and the water vapor pressure expands the pores and cracks of the coal with increasing heat [6].

The materials (liquid or solid) in the microwave cavity heats up rapidly under microwave radiation in response to strong interaction with an electric or magnetic field. Microwave ovens are very useful when they need to efficiently transfer energy to the reaction vessel, but only if they meet the following needs [7]:

- The electric field profile must be homogeneous, either by using state stirrers or by rotating the reaction cavity itself.

- The geometry of the reactor must be well designed considering the depth of microwave penetration.
- The temperature and pressure inside the reaction chamber must be controlled for a continuous control of the process parameters.
- The costs of the reactor and spare parts must be considered.
- Safety and leakage of microwaves must be considered.

In previous studies, limited parameters were considered when examining the electric field and heat of coal [8,9]. Therefore, this study attempts to carry out a comprehensive study of the microwave effects on coal, without considering the limitations of previous experiments, by presenting a complete theory of the problem and examining the most important effective parameters. In the final step, the tests were repeated in a new chamber with different dimensions and the results successfully compared.

## 2. Theory

To measure the rate of heat absorption and transfer in a microwave chamber, we need to model the microwave propagation as well as the heat and mass transfer in the chamber.

### 2.1. Symbols

The list of symbols used in this paper is summarized in Table 1.

**Table 1.** Table of symbols.

| Name | Symbol | Name | Symbol | Name | Symbol |
|------|--------|------|--------|------|--------|
| electric field intensity (V/m) | E | electric current density (A/m$^2$) | J | relative permittivity | $\varepsilon_r$ |
| magnetic field intensity (A/m) | H | electric charge density (C/m$^3$) | P | free space wave number | $k_0$ |
| electric flux density (C/m) | D | permeability (H/m) | $\mu$ | conductivity | $\sigma$ |
| magnetic flux density (Wb/m) | B | relative permeability | $\mu_r$ | angular frequency | $\omega$ |
| magnetic current density (V/m$^2$) | M | permittivity (F/m) | $\varepsilon$ | surface normal vector | N |
| density (kg/m$^3$) | $\rho$ | heat flux (W/m$^2$) | q | mass averaged velocity vector (m/s) | u |
| specific heat capacity at constant stress (J/(kg·K)) | $C_p$ | material's conductivity | k | reaction rate for the species (mol/(m$^3$·s)) | R |
| absolute temperature (K) | T | time | t | diffusive flux vector (mol/(m$^2$·s)) | $J_m$ |
| heat source (W/m$^3$) | Q | concentration of species (mol/m$^3$) | c | Cutoff frequency | fc |
| velocity vector of translational motion (m/s) | U | diffusion coefficient (m$^2$/s) | $D_c$ | Speed of light | $C_0$ |
| wavelength | $\lambda$ | moisture conductivity | $k_m$ | dynamic viscosity (kg/m·s) | v |
| Rayleigh number | $Ra_L$ | specific moisture capacity | $C_m$ | mass transfer coefficient | $k_c$ |
| length | L | | | | |

In this research, "$\nabla$" denotes the gradient, "$\nabla.$" indicates divergence and "$\nabla \times$" states the curl operator.

## 2.2. Maxwell Equations

Solving any electromagnetic problem requires solving Maxwell's equations, which consists of four basic equations that link between the electric field E and the magnetic field B. Their differential form is as follows [10–12]:

$$\nabla \times \overline{E}\,(t) = -\frac{\partial \overline{B}}{\partial t} - \overline{M} \tag{1}$$

$$\nabla \times \overline{H}\,(t) = -\frac{\partial \overline{D}}{\partial t} - \overline{J} \tag{2}$$

$$\nabla \cdot \overline{D} = P \tag{3}$$

$$\nabla \cdot \overline{B} = 0 \tag{4}$$

with

$$B = \mu \cdot H \tag{5}$$

$$D = \varepsilon \cdot E \tag{6}$$

Here, $\varepsilon = \varepsilon_0 \cdot \varepsilon_r$ and $\mu = \mu_0 \cdot \mu_r$ stand, respectively, for the permittivity and permeability of the medium of propagation, with $\varepsilon_0 = 8.85 \times 10^{-12}$ F/m and $\mu_0 = 4\pi \times 10^{-7}$ H/m the respective permeability and permittivity of free space [13]. The magnetic and electric current densities (respectively M and J) as well as the electric charge density $\rho$ are the sources of the fields and are the functions of location.

According to (1) to (6), the Maxwell equation governing the electromagnetic waves inside the microwave chamber is as follows [8]:

$$\nabla \times \mu_r^{-1}(\nabla \times E) - k_0^2 \left( \varepsilon_r - \frac{j\sigma}{\omega \varepsilon_0} \right) E = 0 \tag{7}$$

## 2.3. Heat Transfer

Heat transfer in materials can be achieved in different ways. In fluids, heat is often transferred by convection, in which case the movement of the fluid itself transfers heat from one place to another [14]. Conduction is another method to transfer heat, in which there is no movement of the material and it is carried out by the transfer of energy within that material in contact with another material. The third way to transmit energy is by radiation, which involves the absorption or irradiation of electromagnetic waves.

According to Fourier's law of heat transfer, in a continuous environment, the conductive heat flux ($q_{cond}$) is proportional to the temperature gradient:

$$q_{cond} = -k \cdot \nabla T \tag{8}$$

According to Newton's law of cooling, the convective heat flux depends on the temperature difference between the object surface and the environment (the fluid around the object).

$$q_{conv} = hA_s(T_{surface} - T_{fluid}) \tag{9}$$

The heat exchange rate of pure radiation can be expressed as follows:

$$q_{rad} = \alpha \beta A_s(T_{surface} - T_s) \tag{10}$$

where $0 < \alpha < 1$ and $\beta = 5.67 \times 10^{-8} \frac{W}{m^2 K^4}$

The rate of increase of the object temperature in a non-uniform isotropic environment is obtained from the following equation [15,16]:

$$\rho C_P \frac{\partial T}{\partial t} + \rho C_P\, u \cdot \nabla T + \nabla \cdot (q) = Q_{vap} + Q_{MW} \tag{11}$$

where q includes the conduction heat flux and the radiant heat flux:

$$q = q_{rad} + q_{cond} \tag{12}$$

Here, $Q_{vap}$ is the latent heat of evaporation of coal moisture and $Q_{MW}$ is the heat from microwave radiation. The heat entering the specimen consists of two parts: dielectric losses and magnetic losses:

$$Q_{MW} = Q_{rh} + Q_{ml} \tag{13}$$

$Q_{rh}$ is the heat that enters the object due to the losses of the electric field and it is defined as the real part (Re) of the dot product between the electric current density vector and complex conjugate of electric field intensity vector:

$$Q_{rh} = \frac{1}{2} Re\left( \bar{J} \cdot \overline{E^*} \right) \tag{14}$$

while $Q_{ml}$ is the heat that enters the object due to the losses in the magnetic field and it is calculated as follows [8]:

$$Q_{ml} = \frac{1}{2} Re\left( j\omega \, \bar{B} \cdot \overline{H^*} \right) \tag{15}$$

*2.4. Mass Transfer*

Mass transfer examines the transfer of matter due to differences in chemical potential; the net mass movement of a species from a higher concentration area to a lower one is called mass transfer. Therefore, it is essential to have two areas with different chemical structures for mass transfer. The term is commonly used in engineering for physical processes, which include the diffusion of particles and the convection of chemical species in physical systems. Thus, the convection–diffusion equation is used to investigate this phenomenon. According to Fick's law that expresses the diffusion phenomenon, the general form of the convection–diffusion equation is as follows:

$$\frac{\partial c}{\partial t} + \nabla \cdot \bar{j}_m + \bar{u} \cdot \nabla c = R \tag{16}$$

$$\bar{j}_m = -D_c \nabla c \tag{17}$$

$\nabla \cdot \bar{j}_m$ describes diffusion, $\bar{u} \cdot \nabla c$ describes convection (when coupled to fluid flow) and the reaction rate R describes the creation or destruction of the quantity. For example, R describes how the molecule can be created or destroyed by chemical reactions.

The diffusion coefficient is calculated as follows:

$$D_c = \frac{k_m}{\rho C_m} \tag{18}$$

*2.5. System Design*

The cutoff frequency is the minimum frequency of a wave that can propagate in a waveguide. The cutoff frequency depends on the dimensions of the waveguide. For a rectangular waveguide with width w and height h, the $TE_{mn}$ mode cutoff frequency is calculated as follows:

$$f_{C_0, m, n} = \frac{C_0}{2} \sqrt{\left(\frac{m}{w}\right)^2 + \left(\frac{n}{h}\right)^2}, \; m, n = 0, 1, \ldots \tag{19}$$

Dimensions are usually designed in such a way that $TE_{10}$ mode is used in rectangular waveguides [17]. In a cubic cavity with width w, depth d and height h, the resonance frequency is calculated as follows [18]:

$$f_{res} = \frac{C_0}{2} \sqrt{\left(\frac{m}{w}\right)^2 + \left(\frac{n}{h}\right)^2 + \left(\frac{p}{d}\right)^2}, \; m, n, p = 0, 1, 2, \ldots \tag{20}$$

### 2.6. Initial Conditions

The following initial conditions have been considered:

- The initial temperature of all objects is equal to the ambient temperature of 293.15 K;
- The initial electric field is zero;
- The initial moisture concentration of coal is 3394.4 mol/m$^3$ [6];
- TE$_{10}$ mode is the dominant mode for the rectangular waveguide used in the study [17].

### 2.7. Boundary Conditions

The impedance boundary condition is considered for the walls of the microwave chamber and the waveguide. Its governing equation is as follows [8]:

$$\sqrt{\frac{\mu_0 \mu_r}{\epsilon_0 \epsilon_r - j\frac{\sigma}{\omega}}} n \times H + E - (n{\cdot}E)n = 0 \tag{21}$$

We assumed that there is no heat transfer from the glass plate to the specimen and the loss of electric waves in the specimen is the only reason for the increase in specimen temperature, so the boundary condition between the specimen and the glass plate is defined as follows [19]:

$$n{\cdot}(-k\nabla T) = 0 \tag{22}$$

However, natural convection is considered for specimen walls:

$$-n{\cdot}q = \Delta T * \begin{cases} \dfrac{k}{L}\left(0.68 + \dfrac{0.67 Ra_L^{1/4}}{\left(1 + \left(\frac{0.492k}{vC_P}\right)^{\frac{9}{16}}\right)^{\frac{4}{9}}}\right) & \text{if } Ra_L \leq 10^9 \\[3em] \dfrac{k}{L}\left(0.825 + \dfrac{0.387 Ra_L^{\frac{1}{6}}}{\left(1 + \left(\frac{0.492k}{vC_P}\right)^{\frac{9}{16}}\right)^{\frac{8}{27}}}\right)^2 & \text{if } Ra_L > 10^9 \end{cases} \tag{23}$$

Additionally, there is no mass flux at the boundary between the specimen and the plate, so the boundary conduction condition between the specimen and the plate is defined as follows:

$$n{\cdot}\bar{J}_m = 0 \tag{24}$$

The mass flux of other specimen boundaries is defined as follows:

$$n{\cdot}\bar{J}_m = k_c(c_{air} - c) \tag{25}$$

where c denotes moisture concentration.

### 2.8. Assumptions

The following assumptions are considered to simplify the designed problem:

- The microwave only heats the specimen placed inside it and has no effect on the air or the glass container on which the specimen is placed.
- The wall material is assumed to be copper.
- The specimens placed in the microwave oven are isotropic and homogeneous.
- The electrical, magnetic and thermal properties of coal are constant.
- The motion of water molecules in the magnetic and electric field is simplified as a fixed mass transfer.
- The chemical reaction of objects is ignored (the ignition temperature of coal is 360 °C [20]).

## 3. Methodology

In this simulation, using the COMSOL software, the electromagnetic part of the equation is first solved. In other words, Maxwell equations are solved using the physics of electromagnetic waves in their frequency range, and the microwave propagation and the amount of wave loss inside the microwave is obtained. The amount of energy lost is equal to the amount of energy absorbed in the sample The temperature distribution of the specimen and its heat transfer to the environment are obtained using the Fourier equation in the time domain. By calculating the mass transfer in coal, we simulated the changes in coal moisture.

To start the simulation of the device, a microwave device is first defined with the specifications reported in Table 2. The height of the microwave port from its lower surface is equal to 94 mm [6].

**Table 2.** Microwave oven and coal model sizes.

|  | Width (mm) | Depth (mm) | Height (mm) | Radius (mm) |
| --- | --- | --- | --- | --- |
| Microwave oven | 267 | 270 | 188 | - |
| Waveguide | 50 | 78 | 18 | - |
| Glass plate | - | - | 6 | 113.5 |
| Sample | - | - | 60 | 25 |

Figure 1 shows the defined standard cavity and the coal sample inside it. As shown in Figure 2, the bandwidth of the waveguide is defined in the frequency range of 1.92 GHz to 3.84 GHz. According to the rule of thumb, the total number of geometry elements is 77,000 with an average element quality of 70%. If the maximum system frequency of 4 GHz equivalent to the maximum element size of 15 mm in a vacuum is considered, such quality will be achieved in meshing [21]. Figure 3 shows the optimal mesh quality and reliability of the results. The selected material of the system is identified in Table 3.

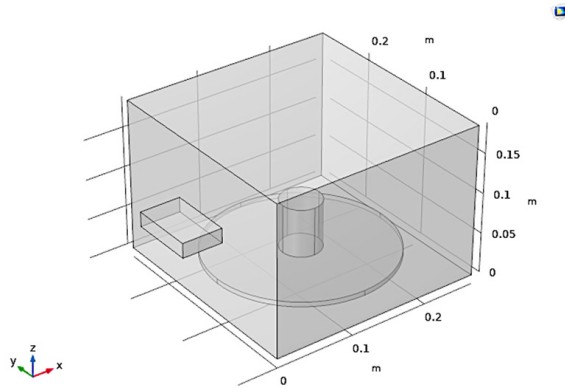

**Figure 1.** Microwave system defined with a cylindrical specimen inside.

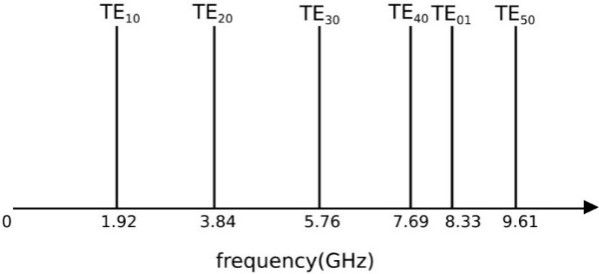

**Figure 2.** Frequency sequence of propagating modes in the used rectangular waveguide according to Table 3.

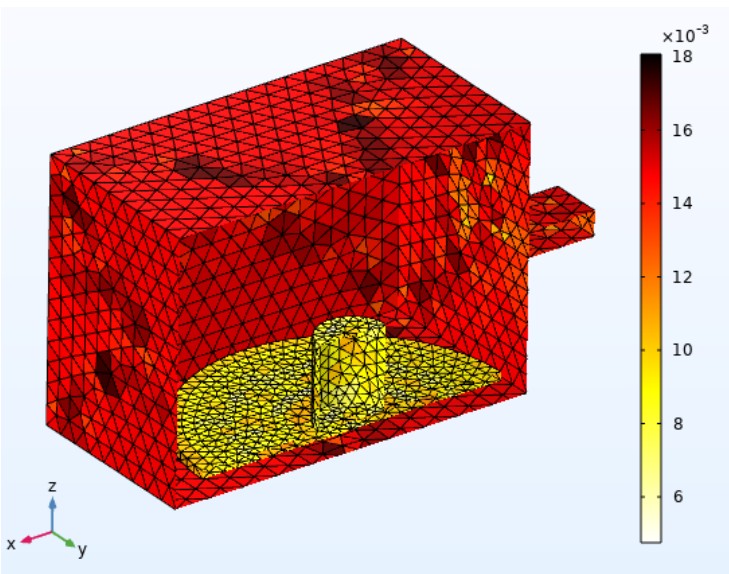

**Figure 3.** System meshing.

**Table 3.** Specifications of materials at 293.15 K [8,22]. In this table, ($\varepsilon'$, $\varepsilon''$) and ($\mu'$, $\mu''$) stand for the (real part, imaginary part) of the relative permittivity and permeability, respectively.

| Name | Units | Values | | | |
|---|---|---|---|---|---|
| Material | - | Coal | Glass | Copper | Air |
| Relative permittivity ($\varepsilon'$-j$\varepsilon''$) | - | Variable | 2.55 | 1 | 1 |
| Relative permeability ($\mu'$-j$\mu''$) | - | 1 | 1 | 1 | 1 |
| Electrical conductivity | S/m | 0.02 | 0 | $5.998 \times 10^7$ | 0 |
| Thermal conductivity | W/(m·k) | 0.478 | - | 400 | 0.0256 |
| Density | Kg/m$^3$ | 1300 | - | 8960 | 1.204 |
| Heat capacity at constant pressure | J/(kg·K) | 4186.8 | - | 385 | 1015.1 |

## 4. Tests and Simulations

In this study, the effects of six cases of coal moisture capacity, operation frequency, input power and energy, specimen location in the microwave and the size and placement of the input waveguide on the intensity and distribution of the electric and thermal fields of coal were investigated. Generally, the applied power and operating frequency are, respectively, 1 kW and 2.45 GHz in the tests, the coal specimens with a moisture capacity of 4.7% are placed in the microwave for 300 s, and each parameter is changed one at a time while the others are kept fixed. Base conditions (Figure 4) have been implemented in this article and also one of the effective parameters has been studied in each experiment to make the results comparable and reliable. In the purpose of this study is to investigate the type of radiation and its effect on the permeability of electromagnetic fields in coal to create the appropriate thermal stress to create cracks and increase the permeability of coal. Therefore, the parameters affecting the distribution of electromagnetic fields and heat in the coal, including the amount of coal moisture (coal type), operating frequency, input power and heating time, location of the waveguide, the size of the waveguide and the location of the coal will be examined. Finally, the parameters are checked again in a secondary standard cavity to separate the parameters related to the size of the environment and the tank from the independent parameters. In Figure 4, the electric and thermal fields inside the chamber and the coal have a moisture capacity of 4.7%, an operating frequency of 2.45 GHz, an input power of 1 kW and a heating time of 300 s.

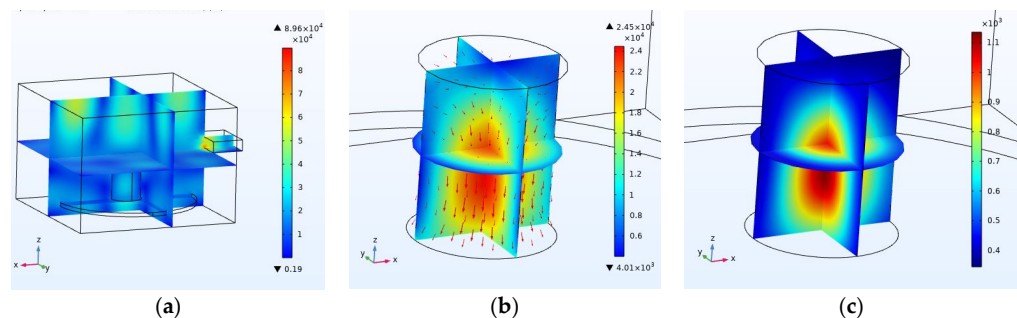

(**a**)    (**b**)    (**c**)

**Figure 4.** (**a**) Electric field inside the chamber. (**b**) Electric field inside the coal. (**c**) Temperature inside the coal [K].

*4.1. Coal Moisture Capacity*

The simulated results in Table 4 show that the moisture capacity of coal has a linear effect on the permittivity of coal. Additionally, due to the constant input power for different specimens, the moisture diffusion coefficient in coal only depends on the moisture capacity of the coal [6].

**Table 4.** Diffusion coefficient and permittivity of coal specimens based on their specific moisture capacity.

| Specific Moisture Capacity $(C_m \%)$ | Real Permittivity $(\varepsilon')$ | Imaginary Permittivity $(\varepsilon'')$ | Diffusion Coefficient $(D\ m^2/s)$ |
|---|---|---|---|
| 0.5 | 0.578 | 0.161 | $1.47 \times 10^{-2}$ |
| 1 | 0.852 | 0.178 | $7.37 \times 10^{-3}$ |
| 2 | 1.401 | 0.211 | $3.69 \times 10^{-3}$ |
| 3 | 1.949 | 0.245 | $2.46 \times 10^{-3}$ |
| 4 | 2.498 | 0.278 | $1.84 \times 10^{-3}$ |
| 4.7 | 2.882 | 0.301 | $1.57 \times 10^{-3}$ |
| 5 | 3.046 | 0.311 | $1.47 \times 10^{-3}$ |
| 6 | 3.595 | 0.345 | $1.23 \times 10^{-3}$ |
| 7 | 4.143 | 0.378 | $1.05 \times 10^{-3}$ |
| 8 | 4.692 | 0.411 | $9.22 \times 10^{-4}$ |
| 9 | 5.240 | 0.444 | $8.19 \times 10^{-4}$ |
| 10 | 5.789 | 0.478 | $7.37 \times 10^{-4}$ |

Figures 5 and 6 show the effect of increasing coal moisture on the size and diffusion of the electric and thermal fields. As can be seen in these figures, the moisture capacity of coal has a significant effect on the electric field distribution in the cavity, the intensity of the electric and thermal fields in coal; furthermore, the coal with a moisture capacity of 5% has the highest average temperature and the highest temperature increase rate compared to the tested samples.

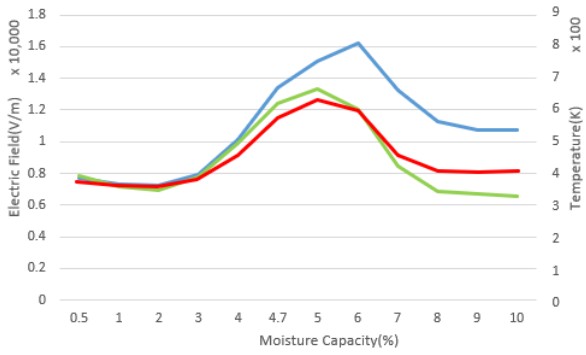

**Figure 5.** In this figure, the green line indicates the average electric field inside the coal, the blue line presents the average electric field inside the entire cavity and the red line denotes the average temperature inside the coal.

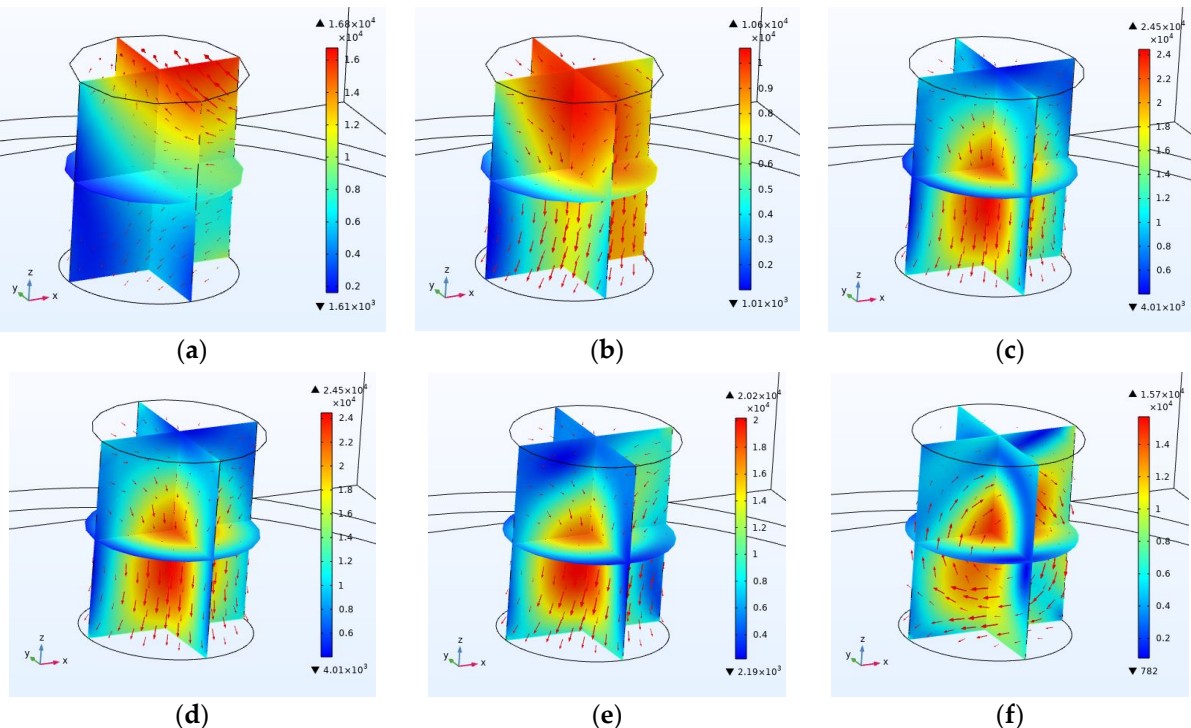

**Figure 6.** Electric field inside the coal with the moisture capacity of (**a**) 0.5%, (**b**) 2%, (**c**) 4.7%, (**d**) 5%, (**e**) 7% and (**f**) 10%. The size of the vectors indicates the size of the electromagnetic field inside the coal and the direction of the vectors indicates the direction of the field at each point inside the coal and how the field is distributed inside the coal.

### 4.2. Effect of Microwave Operating Frequency

Figure 7 shows the effect of frequencies 1.9 to 3.8 GHz applied at a distance of 0.1 GHz on the electric and thermal field inside the coal specimen. According to Figure 7, the frequency of 2.4 GHz has the highest rate of temperature increase and 3 GHz has the lowest rate of temperature increase in coal.

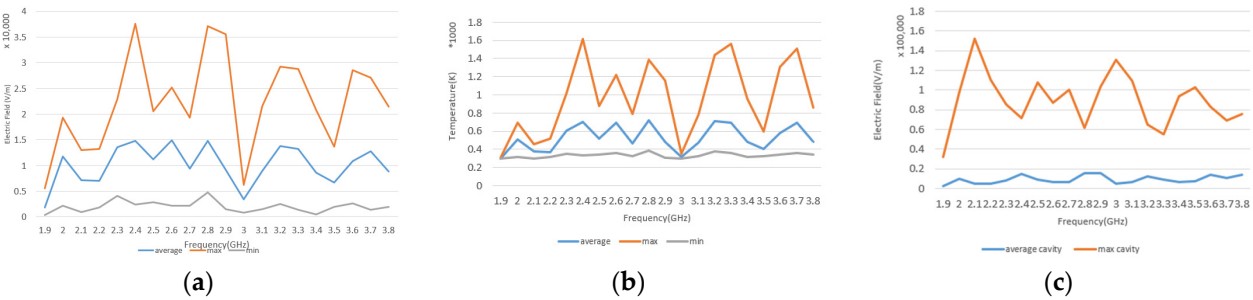

**Figure 7.** (**a**) Average, maximum and minimum electric field in the specimen in the frequency range of 1.9 to 3.8 GHz. (**b**) Average, maximum and minimum temperature in the coal specimen in the frequency range of 1.9 to 3.8 GHz. (**c**) Maximum and average electric field in the cavity.

Additionally, there is the most uniform scattering of electric and thermal fields in coal at a frequency of 2 GHz, and there is the highest density of electric and thermal fields in coal at a frequency of 2.9 GHz.

Figures 8 and 9 show four different frequencies 2, 2.4, 2.9 and 3 GHz.

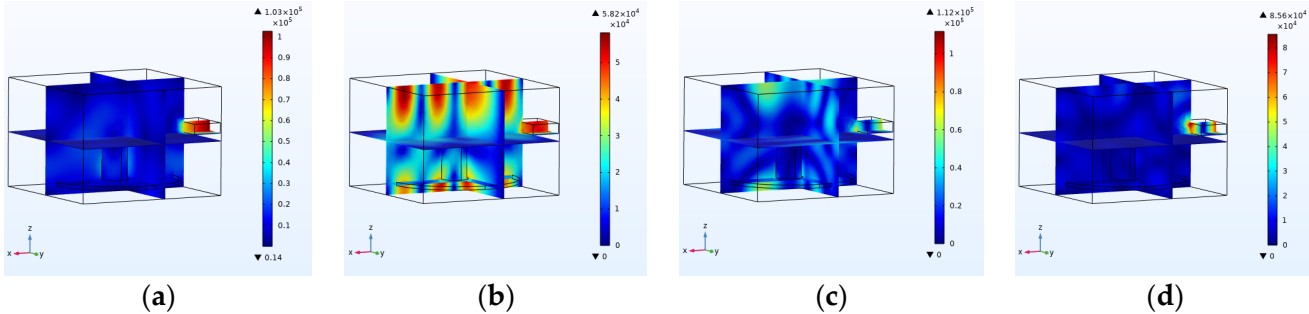

**Figure 8.** Electric field inside the cavity at a frequency of (**a**) 2 GHz, (**b**) 2.4 GHz, (**c**) 2.9 GHz and (**d**) 3 GHz.

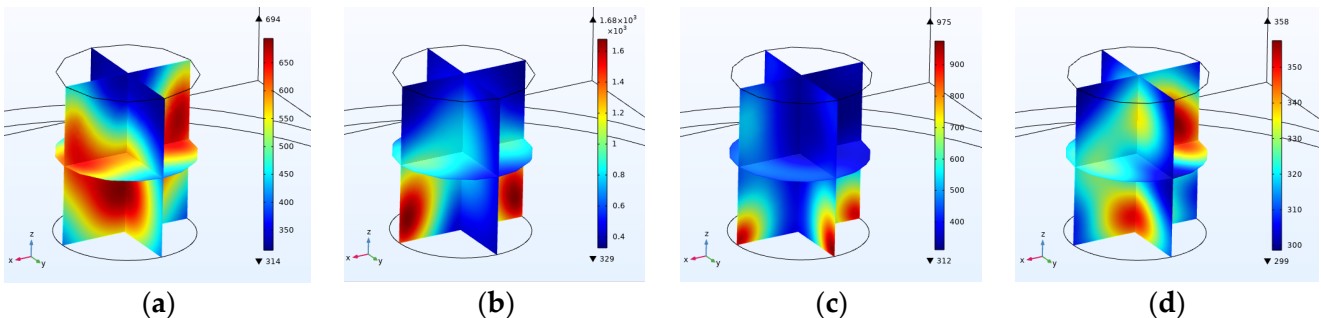

**Figure 9.** Coal temperature at frequency of (**a**) 2 GHz, (**b**) 2.4 GHz, (**c**) 2.9 GHz and (**d**) 3 GHz.

### 4.3. Effect of the Microwave Power and Heating Time

This section addresses the effect of input power on the electric and thermal fields of the system. According to the following relationship, the input power affects moisture conductivity and mass transfer coefficient, which are the effective parameters in mass transfer inside the specimen [6].

$$k_m = 0.01084 \times \text{Power} - 1.25 \tag{26}$$

$$k_c = 10.7194 \times k_m \tag{27}$$

To make the specimens thermally comparable, the input energy is considered constant and equal to 300 kJ (Table 5). According to Figures 10 and 11, the electric field distribution does not depend on the amount of input power and only the size of the electric field increases in proportion to the square root of the input power [23]. The input power also affects the rate of increase in coal temperature by affecting the size of the electric field, and despite the constant input energy to the system, the final temperature of coal with an input power of 3 kW is higher.

**Table 5.** Input power and heating time of coal (The input energy is 300 kJ in all tests).

| Parameter | Unit | Value | | | | | |
|---|---|---|---|---|---|---|---|
| Power | W | 500 | 1000 | 1500 | 2000 | 2500 | 3000 |
| Time | s | 600 | 300 | 200 | 150 | 120 | 100 |

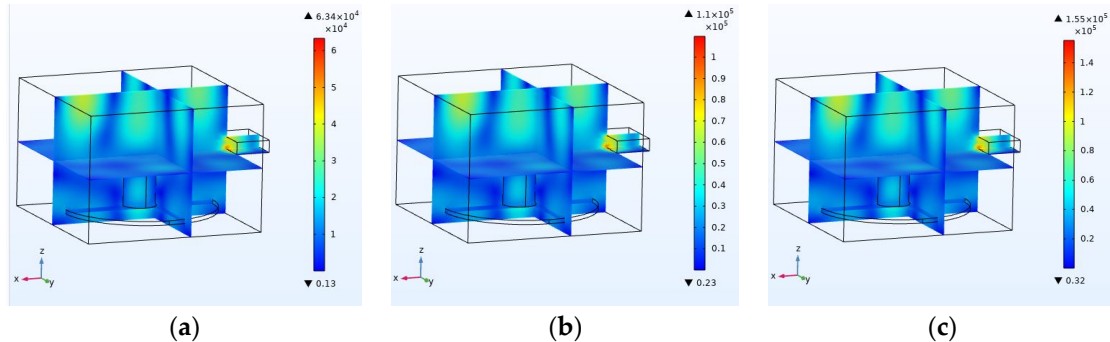

**Figure 10.** Electric field inside the cavity with an input port power of (**a**) 500 watts, (**b**) 1500 watts and (**c**) 3000 watts.

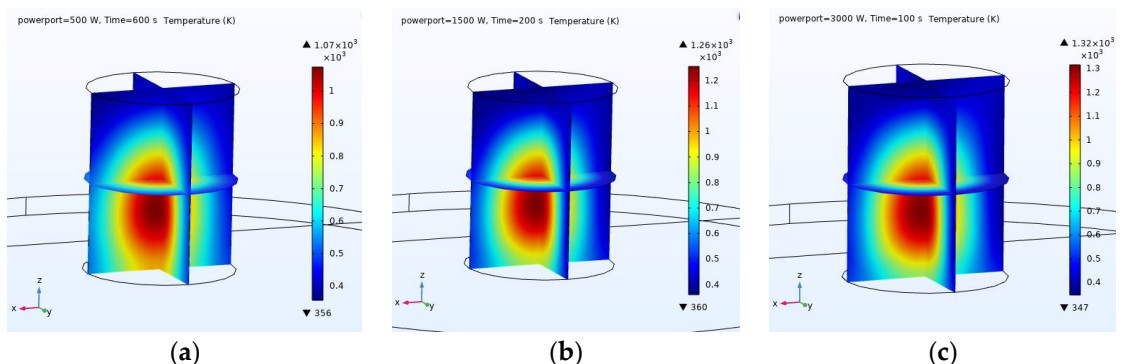

**Figure 11.** Temperature inside the coal with input port power of (**a**) 500 watts, (**b**) 1500 watts and (**c**) 3000 watts.

### 4.4. Effect of the Waveguide Port Location

In this section, the height of the waveguide port, which was set to 0.094 m in the previous experiments, varied. Figure 12 shows the effect of increasing the height of the microwave port on the maximum and average electric field of the cavity and coal. Figure 13 also shows the distribution of electric and thermal fields at the height of 0.017, 0.034, 0.102 and 0.136 m. The height of 0.136 has the highest rate of temperature increase and height 0.034 m has the lowest rate of temperature increase and leads to the most uniform distribution of the electric and thermal fields inside the coal.

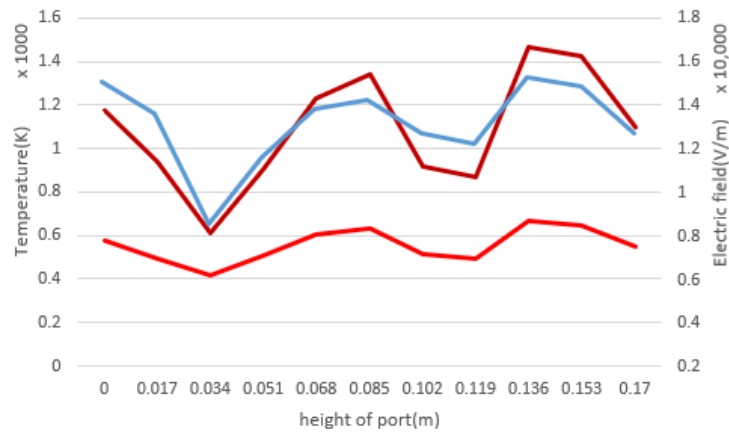

**Figure 12.** The effect of increasing the height of the microwave port with a step of 0.017 m on the average electric field inside the cavity (blue), maximum coal temperature (brown) and average coal temperature (red).

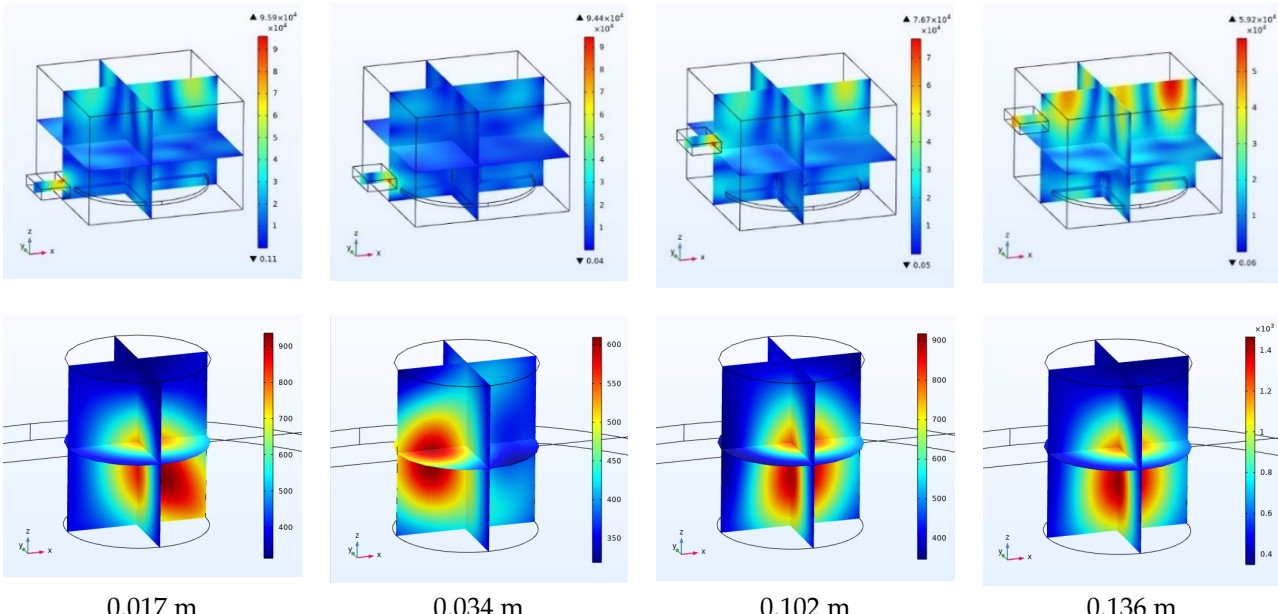

0.017 m      0.034 m      0.102 m      0.136 m

**Figure 13.** Electric field inside the chamber and temperature inside the coal due to the change in the height of the waveguide port.

### 4.5. Effect of the Waveguide Chamber Size

The dimensions of the waveguide were varied one at a time to investigate the effect of the waveguide size. The dimensions of the waveguide have been changed according to Table 6.

**Table 6.** Five different waveguides with different dimensions and sizes.

| Waveguide | Width (m) | Height (m) | Depth (m) |
|:---:|:---:|:---:|:---:|
| 1 | 0.078 | 0.018 | 0.05 |
| 2 | 0.078 | 0.039 | 0.05 |
| 3 | 0.078 | 0.018 | 0.1 |
| 4 | 0.064 | 0.018 | 0.05 |
| 5 | 0.091 | 0.018 | 0.05 |

According to Figures 14 and 15, the distribution of the electric field inside the cavity does not change significantly by changing the dimensions of the waveguide. However, because the waveguide width and height affect the cutoff frequency of the waveguide, the waveguide acts as a high pass filter [24], and the length (depth) of the waveguide only affects the filtering of frequencies below the cutoff frequency. Moreover, when the cutoff frequency of the waveguide is close to the operating frequency (2.45 GHz), the value of the electric field in the cavity and coal and, consequently, the rate of temperature increases inside the coal increases. As can be seen in Figure 16, the electric field inside the cavity and the average temperature inside the coal decrease.

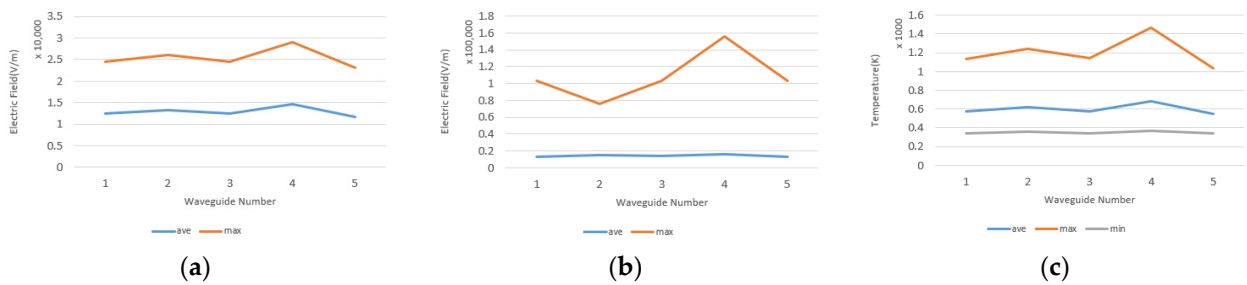

**Figure 14.** (**a**) Average, maximum and minimum electric field in the coal. (**b**) Electric field in the whole cavity. (**c**) Average, maximum and minimum heat field in the coal.

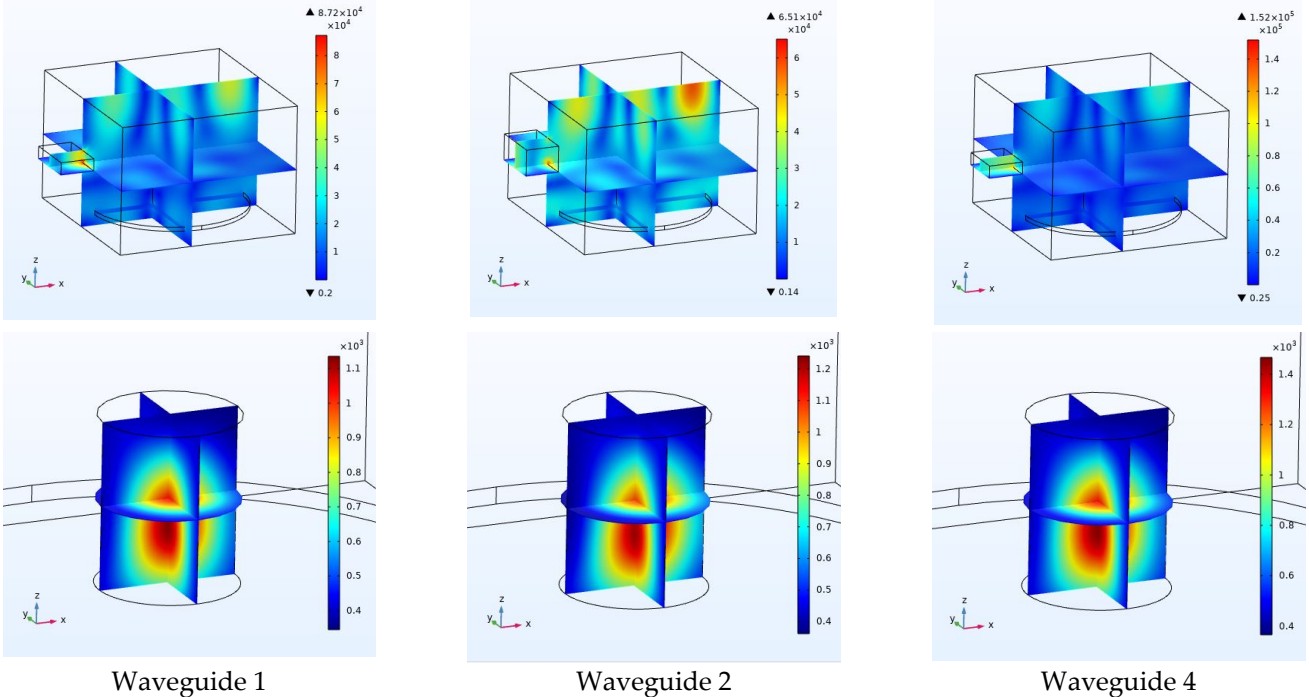

**Figure 15.** Electric field of the cavity with waveguides No. 1, 2 and 4 and heat field in the coal corresponding to waveguides No. 1, 2 and 4.

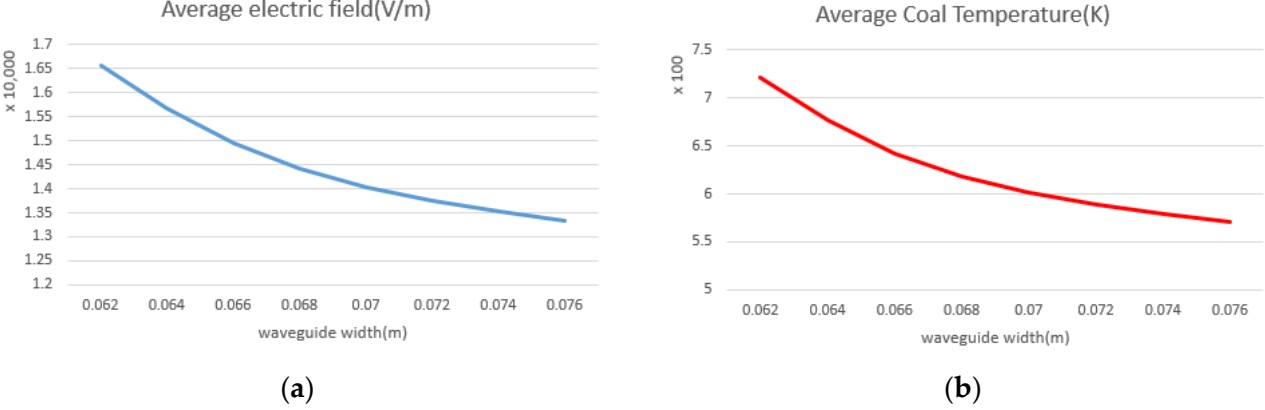

**Figure 16.** (**a**) Average electric field inside the chamber relative vs. waveguide width. (**b**) Average temperature inside the coal vs. waveguide width.

*4.6. Place of Coal*

Another factor that can affect the distribution of electric field and heat inside the chamber is the location of the coal inside it. The average magnitude of the electric field for different heights of the coal bottom (z) and distances of the coal from the waveguide (x) are reported in Table 7.

**Table 7.** Average electric field in terms of coal displacement.

|  | z1 = 0.022 | z2 = 0.042 | z3 = 0.062 | z4 = 0.082 | z5 = 0.102 | z6 = 0.122 |
|---|---|---|---|---|---|---|
| x1 = 0.03 m | 8871.3 | 8972.2 | 8216.8 | 5301.3 | 5633 | 10,621 |
| x2 = 0.1335 m | 13,468 | 7576.1 | 7252.4 | 14,502 | 8077.2 | 7613 |
| x3 = 0.237 m | 10,248 | 10,938 | 11,740 | 8768.7 | 11,305 | 17,366 |

According to Figure 17, the highest average temperature, the most uniform thermal field and the densest thermal field have been obtained for (x2, z4), (x2, z2) and (x1, z3), respectively.

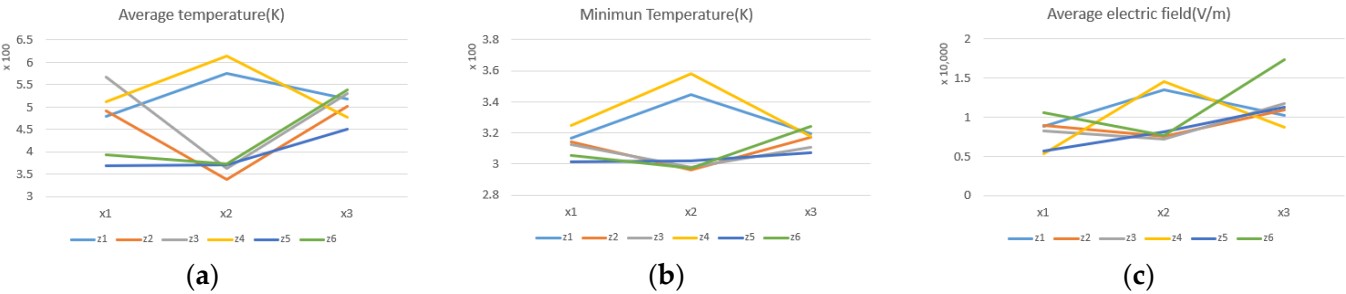

(a)        (b)        (c)

**Figure 17.** (**a**) Average temperature inside the coal. (**b**) Minimum temperature inside the coal. (**c**) Average electric field inside the cavity.

Among the six factors mentioned, i.e., coal moisture capacity, frequency, input power, waveguide location, waveguide dimensions and coal location, the dimensions of the chamber are one of the most important factors affecting the distribution of the electric and thermal fields. In fact, the chamber walls form the boundary conditions of the system and affect the resonant frequency of the chamber. Therefore, the tests performed in the next part were repeated in a chamber with different dimensions to validate the results.

## 5. Changing Cavity Size

The dimensions of the resonance chamber are one of the most basic parameters affecting the resonance frequencies and consequently, the electric and thermal field distribution inside the chamber and coal [18]. Therefore, to demonstrate the reliability of the obtained results, it is necessary to consider a chamber with different dimensions. The dimensions of the new chamber are 296 × 366 × 302 mm [18]. The coal, with moisture capacity of 4.7%, input port power of 1000 watts and operating frequency of 2.45 GHz, was placed in the chamber for 300 s. The dimensions of the waveguide, glass plate and coal are summarized in Table 3. The height of the waveguide port from the bottom of the chamber is 0.021 m so that the electric and thermal field of the coal is the same as in the previous chamber (Figure 4). The structure of the new chamber is shown in Figure 18.

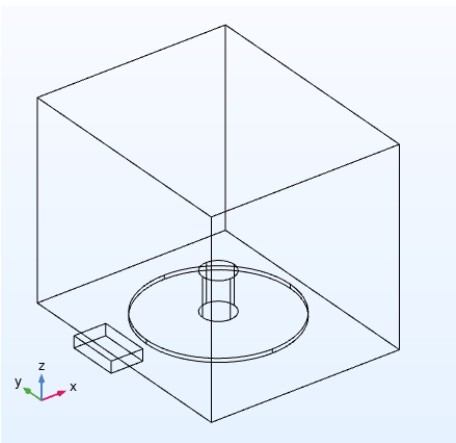

**Figure 18.** Coal placement in the second chamber.

The electric and thermal fields inside the chamber and the coal are shown in Figure 19.

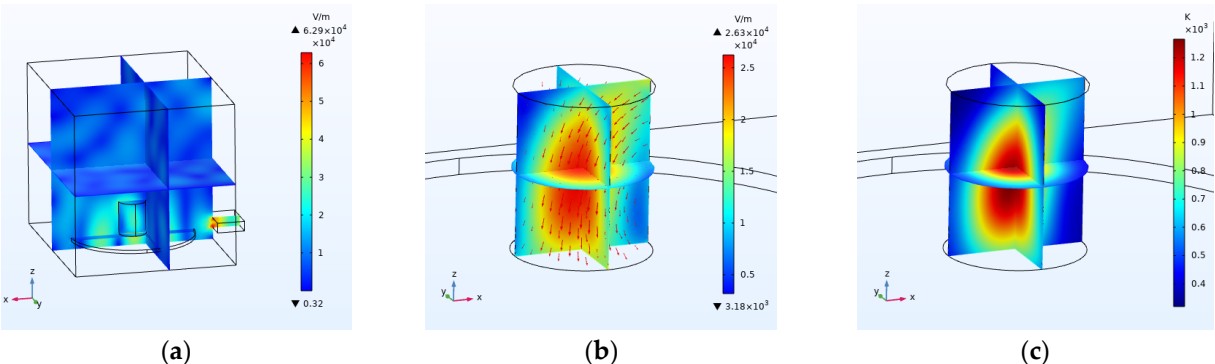

(a)                                    (b)                                    (c)

**Figure 19.** (**a**) Electric field inside the chamber. (**b**) Electric field inside the coal. (**c**) Temperature inside the coal (K).

The parameters were changed one at a time following the same procedure as in the previous section.

*5.1. Coal Moisture Capacity*

Specific moisture capacity, diffusion coefficient and permittivity of the coals were set according to Table 4. Figures 20 and 21 show the effect of increasing coal moisture on the electric and thermal field distribution. As in the previous chamber, a moisture of 5% leads to the highest temperature increase in the coals.

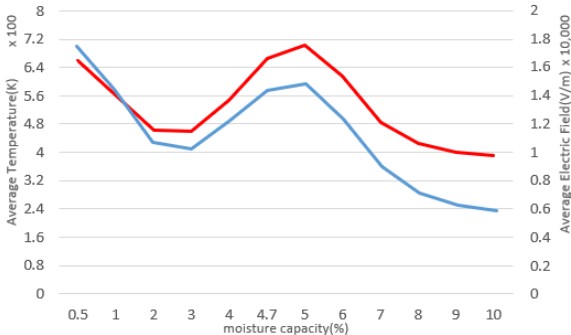

**Figure 20.** Blue indicates the average electric field inside the coal and red indicates the average temperature inside the coal.

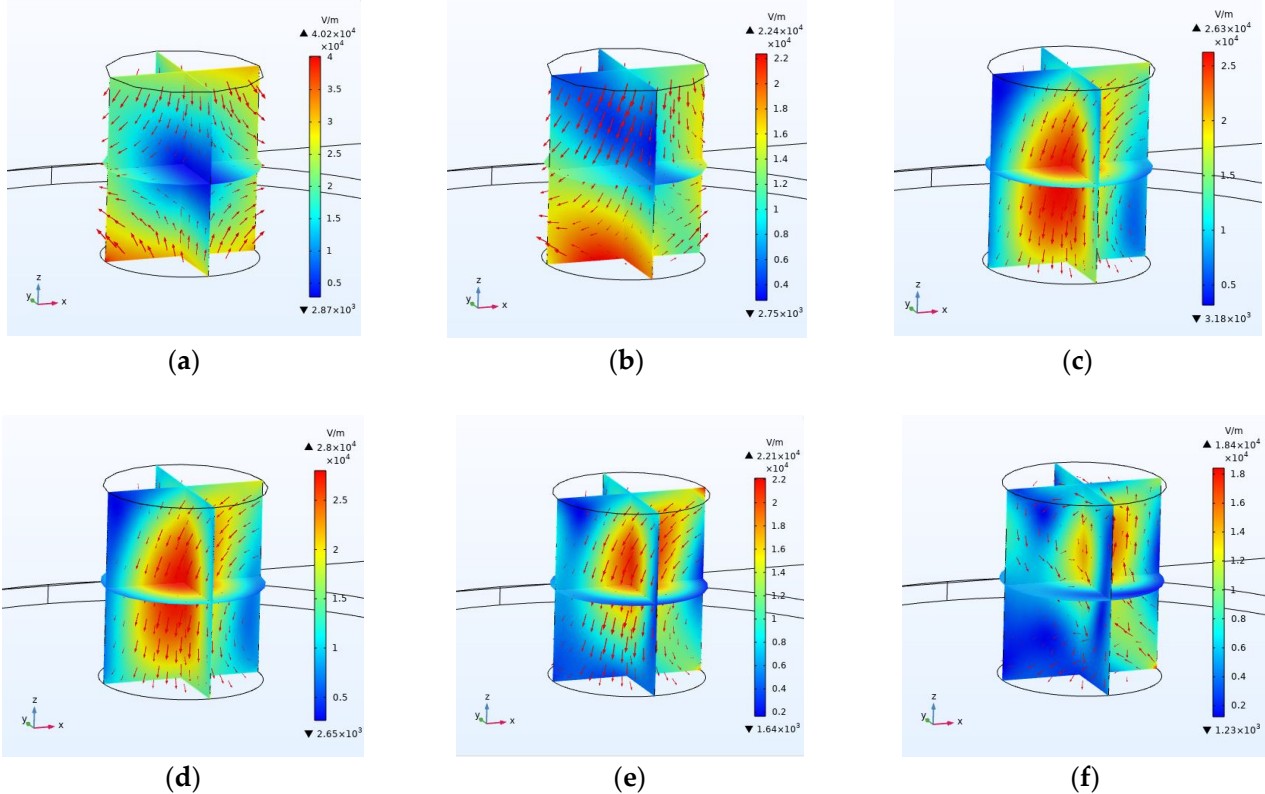

**Figure 21.** Electric field inside the coal with the moisture capacity of (**a**) 0.5%, (**b**) 2%, (**c**) 4.7%, (**d**) 5%, (**e**) 7% and (**f**) 10%. The size of the vectors indicates the size of the electromagnetic field inside the coal and the direction of the vectors indicates the direction of the field at each point inside the coal and how the field is distributed inside the coal.

*5.2. Microwave Frequency*

Figure 22 shows the effect of varying frequency from 1.9 to 3.8 GHz (with a distance of 0.1) on the electric and thermal fields inside the coal specimen. Uniform electric and thermal fields in coal frequency were obtained for 2.1 GHz while the highest temperature increase in coal was shown for a frequency of 3.5 GHz and the lowest temperature increase for 2.2 GHz. Note that these values are different from those obtained in Section 5.2.

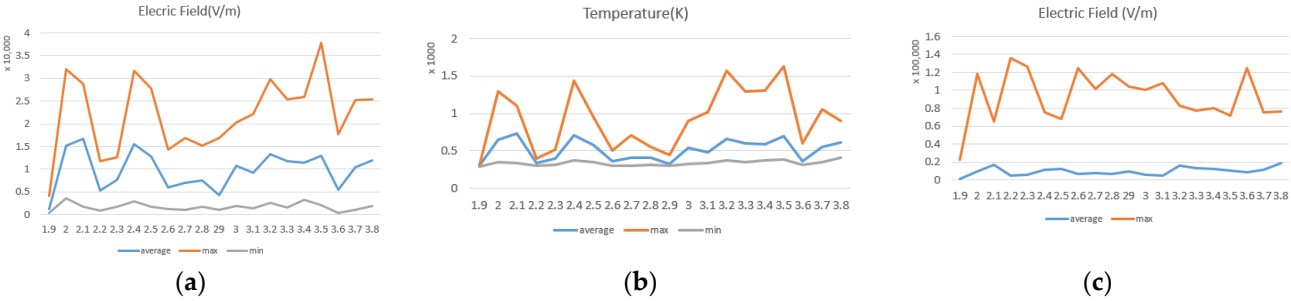

**Figure 22.** (**a**) Average, maximum and minimum electric field inside the specimen in the frequency range of 1.9 to 3.8 GHz. (**b**) Average, maximum and minimum temperature in the coal specimen in the frequency range of 1.9 to 3.8 GHz. (**c**) Maximum and average electric field inside the cavity.

Figures 23 and 24 also show four frequency examples.

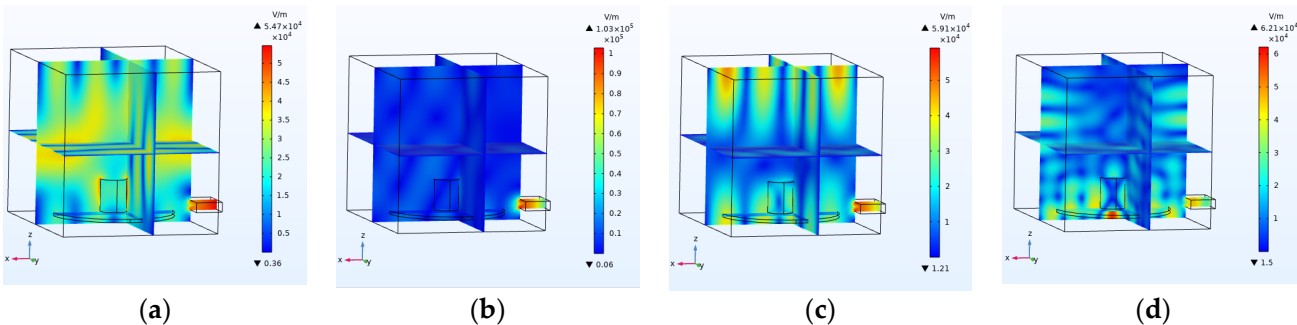

**Figure 23.** Electric field inside the cavity at the frequency of (**a**) 2.1 GHz, (**b**) 2.2 GHz, (**c**) 2.4 GHz and (**d**) 3.5 GHz.

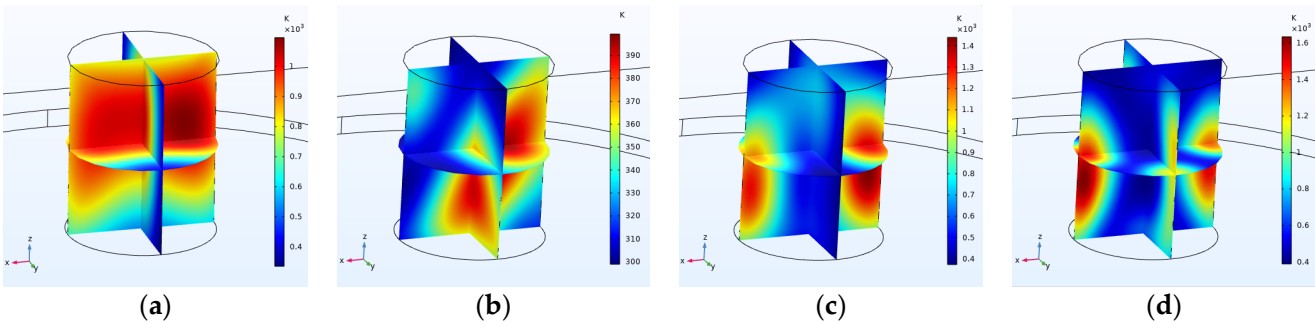

**Figure 24.** Coal temperature at the frequency of (**a**) 2.1 GHz, (**b**) 2.2 GHz, (**c**) 2.4 GHz and (**d**) 3.5 GHz.

*5.3. Microwave Time and Power*

This section investigates the effect of input power on the electric field and heat of the system. To make the specimens comparable in terms of heat, the input energy was assumed constant and equal to 300 kJ. The input power was set according to Table 5 and the results compared. According to Figures 25 and 26, the electric field distribution does not depend on the amount of input power, and only the electric field size increases in proportion to the square root of the input power, which confirms the results obtained with the smaller cavity (Section 4.3).

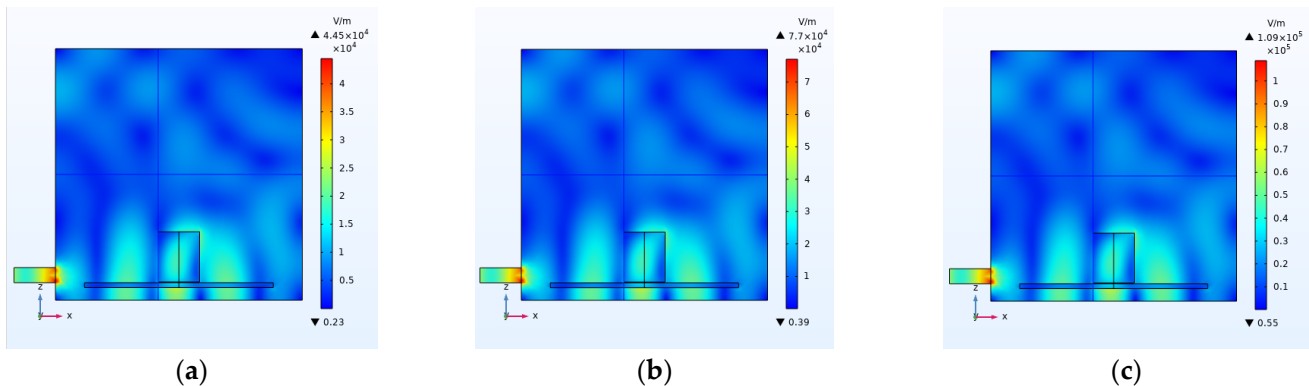

**Figure 25.** Electric field inside the cavity with the input port power of (**a**) 500 watts, (**b**) 1500 watts and (**c**) 3000 watts.

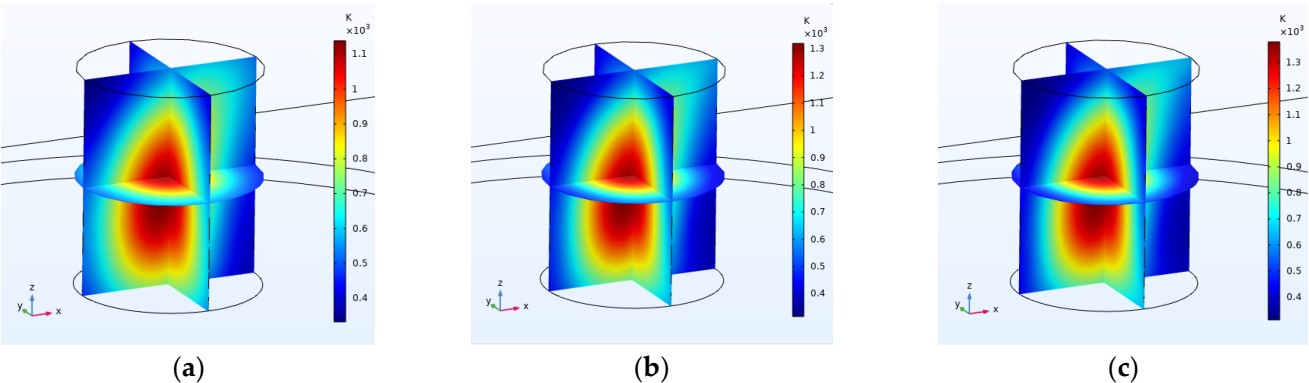

**Figure 26.** Temperature inside the coal with the input port power of (**a**) 500 watts, (**b**) 1500 watts and (**c**) 3000 watts.

### 5.4. Waveguide Place

In this section, the waveguide height is varied. Figure 27 shows the effect of increasing the microwave port height on the average electric field of the cavity and the maximum and average thermal field of the coal. The height of 0.051 m gave the lowest temperature increase rate while a height of 0.17 m produced the highest temperature increase rate inside the coal.

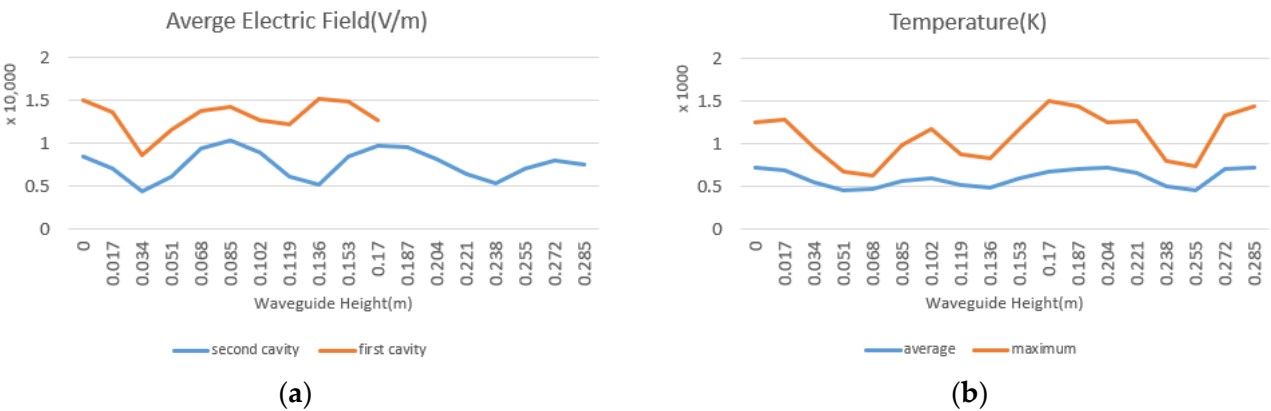

**Figure 27.** (**a**) The effect of increasing the height of the microwave port with a step of 0.017 m on the average electric field inside the cavity and its comparison with the first cavity. (**b**) Maximum and average coal temperature.

Figure 27a indicates that the change in the height of the waveguide causes oscillating changes in the average electric field of both chambers. Figure 28 shows the distribution of electric and thermal fields at the altitudes of 0.017, 0.034, 0.136 and 0.285 m.

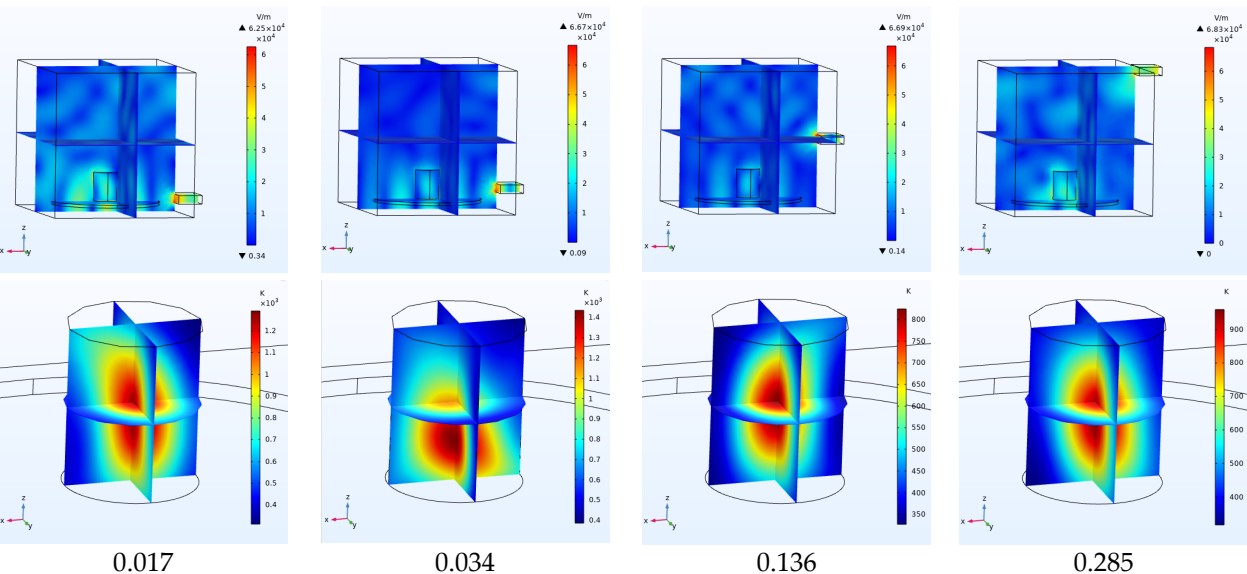

|     |     |     |     |
| --- | --- | --- | --- |
| 0.017 | 0.034 | 0.136 | 0.285 |

**Figure 28.** Electric and thermal fields inside the cavity and coal vs. the height of the waveguide port.

### 5.5. Waveguide Size

The dimensions of the waveguide were changed one at a time to investigate the effect of waveguide size (Table 8). As can be seen in Figures 29 and 30, by changing the dimensions of the waveguide, the distribution of the field inside the cavity does not change significantly. However, when the cutoff frequency of the waveguide is close to the operating frequency (2.45 GHz), the size of the electric field in the cavity and the coal and, consequently, the size of the thermal field inside the coal increase, similarly to what we obtained with the previous cavity (Section 4.5). Figure 31 shows that with increasing wavelength, the average electric field inside the cavity and the average temperature inside the coal decrease.

**Table 8.** Dimensions of the tested waveguides.

| Waveguide | Width (m) | Height (m) | Depth (m) |
| --- | --- | --- | --- |
| 1 | 0.078 | 0.018 | 0.05 |
| 2 | 0.078 | 0.039 | 0.05 |
| 3 | 0.078 | 0.018 | 0.1 |
| 4 | 0.064 | 0.018 | 0.05 |
| 5 | 0.091 | 0.018 | 0.05 |

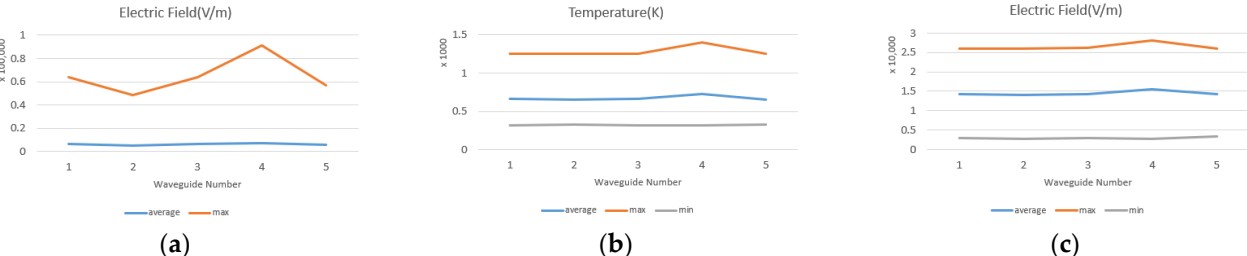

|     |     |     |
| --- | --- | --- |
| (**a**) | (**b**) | (**c**) |

**Figure 29.** (**a**) Average, maximum and minimum electric field in the coal. (**b**) Maximum and average electric field in the whole cavity. (**c**) Average, maximum and minimum thermal field in the coal.

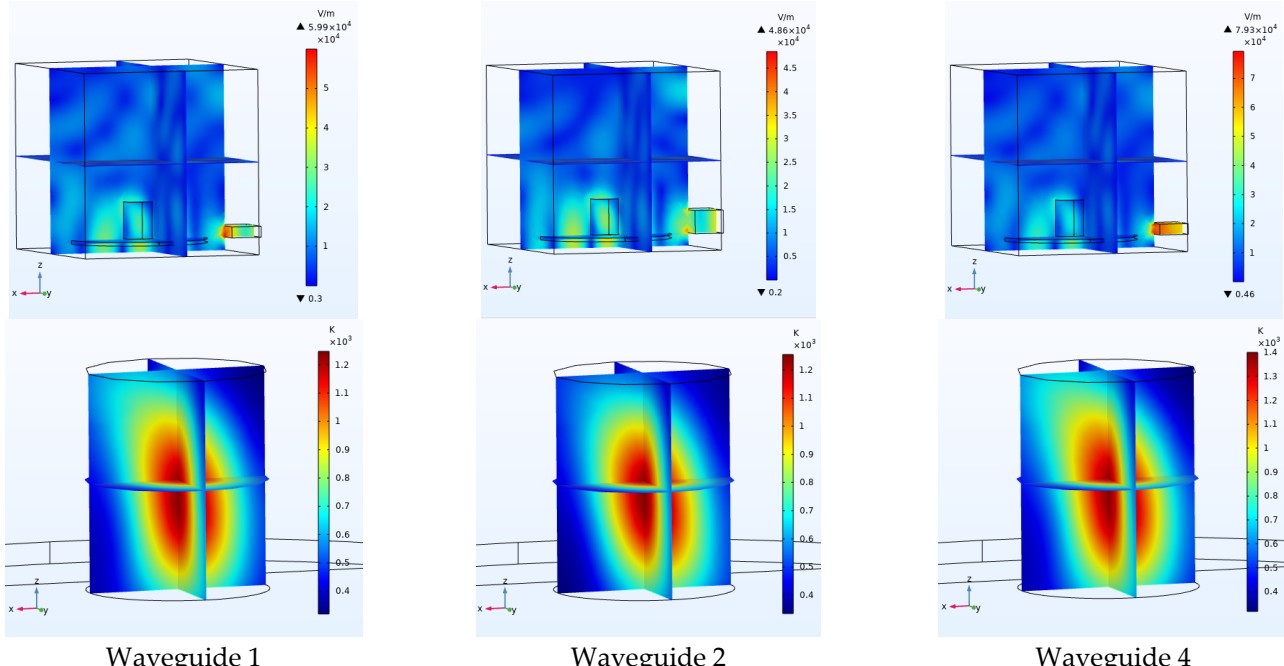

Waveguide 1          Waveguide 2          Waveguide 4

**Figure 30.** Electric field of the cavity with waveguide No. 1, 2 and 4 and thermal field in the coal corresponding to waveguide No. 1, 2 and 4.

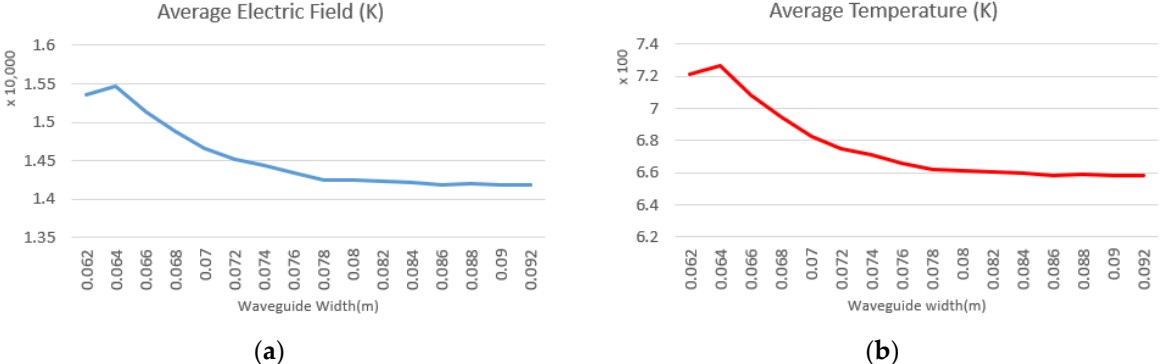

(**a**)              (**b**)

**Figure 31.** (**a**) The average electric field inside the chamber and (**b**) the average coal temperature that decrease with increasing waveguide width.

*5.6. Place of Coal*

Another factor that affects the electric and thermal field distribution inside the chamber is the place of the coal inside it. According to Table 9, the height of the coal bottom (z) and the distance of the coal from the waveguide (x) varied. According to Figure 32, the place of coal $(x_3, z_6)$ had the highest average temperature rise and the densest temperature occurred at $(x_1, z_1)$, i.e., similar to in the previous chamber, while the highest temperature rise and highest thermal field density happened when the coal was attached to the end of the waveguide. As in the previous chamber, the highest temperature rise and the highest thermal field density occurred when the coal was attached to the waveguide aperture, but no clear relationship was observed between the place of the coal and the electric and thermal field distribution inside the coal.

**Table 9.** Maximum electric field in terms of coal displacement.

|  | z1 = 0.022 | z2 = 0.042 | z3 = 0.062 | z4 = 0.082 | z5 = 0.102 | z6 = 0.122 |
|---|---|---|---|---|---|---|
| x1 = 0.03 m | 48,921 | 17,035 | 12,489 | 17,755 | 18,440 | 12,992 |
| x2 = 0.1485 m | 25,998 | 15,330 | 16,072 | 17,902 | 14,317 | 14,453 |
| x3 = 0.267 m | 22,924 | 18,672 | 13,708 | 15,357 | 28,802 | 23,911 |

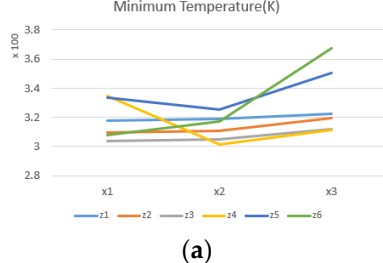

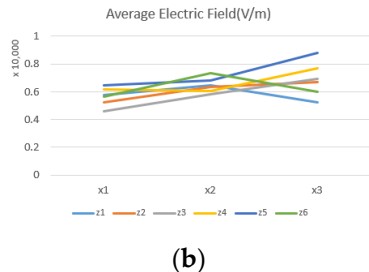

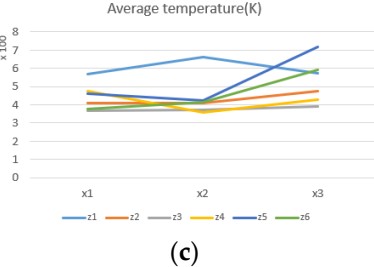

(**a**)            (**b**)            (**c**)

**Figure 32.** (**a**) Average temperature inside the coal. (**b**) Minimum temperature inside the coal. (**c**) Average electric field inside the cavity.

A comparison between the tests performed in the chamber indicates that the moisture of 5%, the highest input power (3000 watts) and a cutoff frequency close to the operating frequency cause the highest average temperature inside the coal. However, the appropriate operating frequency, the right waveguide location and the suitable coal place vary depending on the size of the chamber.

## 6. Conclusions

Methane gas in coal is one of the most hazardous flammable and explosive agents in coal mines. Therefore, to prevent explosions, risk of methane accumulation and to prevent environmental pollution, it is necessary to extract methane gas during coal mining operations. Using microwave radiation is one of the newest methods of extracting methane gas in coal. For this purpose, this paper examined the parameters affecting the electric and thermal field distribution in both the coal and the microwave chamber with the aim to obtain suitable conditions for the extraction of methane gas in coal. The results show that the moisture of 5%, the highest input power and cutoff frequency close to the operating frequency cause the highest average temperature inside the coal, but many parameters such as operating frequency, waveguide location and coal location should be selected depending on the chamber size. Therefore, the most important parameter affecting the electric and thermal fields of coal is the dimensions of the resonant chamber.

**Author Contributions:** Conceptualization, A.J.; methodology, A.J. and A.M.; validation, A.J. and A.M.; formal analysis, A.J. and A.M.; investigation, A.J., A.M. and R.A.; writing—original draft preparation, A.J. and A.M.; writing—review and editing, A.J., A.M. and R.A.; visualization, A.J., A.M. All authors have read and agreed to the published version of the manuscript.

**Funding:** We express our gratitude to the Minister of Economic Development, Trade, and Tourism for funding this project through Major Innovation Funds. The authors also would like to acknowledge the NSERC (Grant Nos. NSERC RGPIN-2017-04516 and NSERC CRDPJ 537378-18) for further funding of this project.

**Institutional Review Board Statement:** Not applicable.

**Informed Consent Statement:** Not applicable.

**Data Availability Statement:** All data are shown in this paper.

**Conflicts of Interest:** The authors declare no conflict of interest. The funders had no role in the design of the study; in the collection, analyses, or interpretation of data; in the writing of the man- uscript; or in the decision to publish the results.

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
