# Peer review of "Numerical Simulation and Optimization of Microwave Heating Effect on Coal Seam Permeability Enhancement"

_technologies, doi:10.3390/technologies10030070_

Round 1

Reviewer 1 Report

The paper is nice and well written but is not under the MDPI template. It also needs minor changes.

Figures 6 and 21 should indicate in the caption what the small arrows in the figure mean.

Figures 7, 14, 22 and 29 should have larger letters and also in the yy axius, the scale should be in potencies of 10. Axes also should have labels.

Figure 16a - waveguide is mispelled in the figure.

Figures 17, 27, 31 and 32 have no axes labels.

Author Response

Dear Editors and Reviewers,

We are very grateful to all of you for taking the time to carefully evaluate our paper “Numerical simulation and optimization of microwave heating effect on coal seam permeability enhancement” (Manuscript ID: technologies-1695797) and for the insightful comments and suggestions. In light of the review feedback, we have carefully revised the manuscript and believe that it is improved as a result of the modifications.

On the following pages, we have included the comments (in blue) from each of the reviewers, our response to all the questions (in black), and documented resulting changes to the manuscript (in yellow). Also attached is the modified version of the manuscript.

As Covid-19 tears its way through communities across the globe, the normal scientific research is seriously affected.We wish you and your family health and safety during the coronavirus outbreak.

Best regards,

All the authors.

Reviewer 2 Report

This is a very interesting paper where the authors look into the impact of microwave radiation on the different electric and thermal field distribution within coal. 

Detailed comments are included in the annotated paper. However, I will include some generic comments here. 

  1. The title of the paper talks about "coal seam permeability enhancement". However, the effect of the electric and thermal field distribution within coal on the permeability seems to be missing. In the conclusion, there should be sort of relationship that needs to be defined.
  2. There are a lot of images included in the paper some of which has results which are different from the rest of the group. It would be interesting if the authors explains why those discrepancies were obtained. All it says is, this is better or something like that. But the why is missing. 
  3. What about the effect of the size of coal samples? Was that studied here or are there previous studies to show any impact? That would be really interesting to see. 
  4. The conclusion needs to be a bit more comprehensive. 

Author Response

(The authors gave the same response as above.)

Reviewer 3 Report

The paper called Numerical simulation and optimization of microwave heating effect on coal seam permeability enhancement by Ali Jebelli, Arezoo Mahabadi, Rafiq Ahmad. The paper is very good and interesting. The main idea is clearly explained and I’m really impressed by the variety of research methods.

There are some major aspects I would like to highlight.

  • It would be necessary to emphasize what is the main scope and purpose of the presented publication, it is difficult to find it in the text
  • In the abstract presented, the importance of the publication should be more concisely described, including more extensive consideration of the methods, analyses and results of the research
  • The publication is very well described in terms of theoretical.
  • Rather, the publication reflects the nature of the engineering work.
  • What is the future direction of development?
  • The title of the publication would be worth rewording,
  • Methodologies could be more clearly explained in some parts of the paper.
  • What this work contributes to science.

The presented conclusions may be of fundamental importance, therefore they should be presented in a better light and the author’s should emphasize the original research contribution. I believe, that suggested amendments will significantly increase the relevance of the publication and will improve it. After applying all required changes, the paper is suitable for publication.

Author Response

(The authors gave the same response as above.)

Round 2

Reviewer 2 Report

I don't see any of my comments addressed in the paper. 

Reviewer 3 Report

Thank you for the changes made.

 Accept in present form